# Dietary Interventions in Ulcerative Colitis: A Systematic Review of the Evidence with Meta-Analysis

**DOI:** 10.3390/nu15194194

**Published:** 2023-09-28

**Authors:** Marta Herrador-López, Rafael Martín-Masot, Víctor Manuel Navas-López

**Affiliations:** 1Pediatric Gastroenterology and Nutrition Unit, Hospital Regional Universitario de Málaga, 29011 Málaga, Spain; herradorlopezm@gmail.com (M.H.-L.); victor.navas@gmail.com (V.M.N.-L.); 2The Biomedical Research Institute of Málaga (IBIMA), 29010 Málaga, Spain

**Keywords:** ulcerative colitis, diet, children, inflammatory bowel disease, exclusion diet, Crohn’s disease

## Abstract

(1) Background: Ulcerative colitis (UC) is a chronic colon inflammation caused by genetic and environmental factors, including diet. This systematic review and meta-analysis aims to assess the impact of diet on UC management in children and adults (2) Methods: A comprehensive search across databases yielded relevant studies, and risk of bias in randomized controlled trials (RCTs) was assessed using the Cochrane Risk of Bias tool. This study was conducted in conformity to the 2020 PRISMA guidelines. The certainty of evidence for outcomes was evaluated using GRADE methodology. Meta-analysis was performed using Review Manager software version 5.4. (3) Results: Fourteen RCTs were included, results indicated higher clinical response, remission, and endoscopic remission rates in diet-treated groups. Carrageenan-free, anti-inflammatory, and cow milk protein elimination diets showed no significant advantages in maintaining clinical remission. However, a study involving fermented cow milk with bifidobacterial demonstrated favorable outcomes. Overall, pooled analysis leaned in favor of dietary intervention for sustaining clinical remission; (4) Conclusions: The relationship between diet and UC is an evolving terrain that demands deeper exploration. This systematic review and meta-analysis highlight the evolving relationship between diet and UC, necessitating further exploration. While understanding grows, adopting personalized dietary approaches could alleviate symptoms, and support a more positive disease trajectory.

## 1. Introduction

Ulcerative colitis (UC), a chronic inflammatory disorder primarily localized in the colon, is understood as a complex interplay between genetic susceptibility and environmental factors, including diet, that trigger an abnormal immune response [1]. Its incidence is estimated at approximately 15 to 20% of UC cases occurring in childhood. The prevalence of the disease varies geographically, with Europe and North America experiencing higher rates. Over the past decades, there has been a notable increase in UC incidence worldwide. The intricate mechanisms through which dietary components impact UC remain incompletely understood [2]. In terms of indirect effects, it has been suggested that changes in gut microbiota composition induced by diet contribute to the initiation or maintenance of inflammation [3]. High-fat and/or high-sugar diets have been shown to lead to mucosal dysbiosis characterized by increased pro-inflammatory Proteobacteria and decreased anti-inflammatory bacteria [4]. Diet can also affect the metabolic functions of the microbiota. Short-chain fatty acids (SCFAs) derived from bacterial fermentation of non-digestible fiber, such as acetate, propionate, and butyrate, play a role in maintaining mucosal barrier function and modulating immune function [5]. Diets low in fiber, high in sugars and fats, have been linked to reduced SCFAs production, making individuals more susceptible to UC [4].

Direct effects of dietary factors on cells have also been explored. Long-term intake of high-fat or high-sugar diets can lead to the production of excessive free radicals. This initiates a cascade of oxidative stress and inflammatory signaling pathways, resulting in the increased production of inflammatory cytokines and chemokines. This, in turn, triggers an enhanced immune response characterized by the recruitment of immune cells, leading to a further increase in ROS (Reactive Oxygen Species) production catalyzed by NADPH oxidases (NOXs). Concurrently, there is prolonged activation of inflammatory, redox, and apoptosis signaling pathways. Ultimately, this cascade of events culminates in several adverse outcomes: a dysfunction in the immune response, increased cell apoptosis, disrupted mucosal homeostasis, and impaired intestinal barrier function, which can lead to mucosal damage [6].

Micronutrient depletion, such as luminal intestinal iron depletion, can directly impact intestinal epithelial cell and T cell function [7]. Deficiency in zinc can affect intestinal barrier integrity and permeability [8]. Vitamin D has been studied for its role in bolstering the innate immune system and reducing inflammation [9]. The content of *n*-3 polyunsaturated fatty acids (PUFAs), like eicosapentaenoic acid (EPA) and docosahexaenoic acid (DHA), has been associated with lower UC development probability by inhibiting inflammatory genes. Conversely, dietary arachidonic acid (*n*-6 PUFA) increases the likelihood of UC development [10].

Overall, a Western diet rich in processed foods, saturated fats, sugars, red meats, and refined grains is associated with increased mucosal inflammation [11]. Conversely, consumption of fruits and vegetables seems to decrease inflammatory risk in UC [12]. Food additives like maltodextrin and emulsifiers or thickening agents such as carboxymethylcellulose, carrageenan, and xanthan gum can also have detrimental effects on intestinal homeostasis [13]. Despite the growing number of reviews examining the impact of dietary factors on UC development [14,15,16], the current dietary recommendations for managing the disease remain scarce and lack a robust scientific basis [17]. Consequently, the question arises whether dietary treatment can enhance remission induction rates or aid in maintaining remission among UC patients.

This comprehensive systematic review endeavors to scrutinize and synthesize the existing evidence regarding the impact and efficacy of various diets in the management and treatment of UC in both pediatric and adult populations.

## 2. Materials and Methods

### 2.1. Search Strategy and Study Selection

The study was conducted as per the Preferred Reporting Items for Systematic Reviews and Meta-Analyses (PRISMA) statement [18,19], the PRISMA statement [20], and the PRISMA-P guidelines and checklist [18,21].

The search strategy was developed following the PICO methodology, which breaks down as follows: Patient (P): Individuals diagnosed with UC. Intervention (I): Implementation of a specific dietary treatment. Comparison (C): Comparison with the usual treatment of the disease, which does not involve dietary strategies. Outcome (O): Evaluation of whether the dietary treatment improves the rates of induction of clinical, endoscopic, or histological remission, or contributes to the maintenance of remission in patients with UC.

The formulated question was as follows: In individuals diagnosed with UC, does the implementation of a specific dietary treatment compared to the usual treatment that does not include dietary strategies lead to an improvement in the rates of induction of clinical, endoscopic, or histological remission, or favor the maintenance of remission?

### 2.2. Inclusion and Exclusion Criteria

Studies selected for review were confined to RCTs investigating the effects of defined solid or liquid diets, as compared to a control diet, in individuals with UC. The intervention group was mandated to adhere to a rigorously delineated diet, rather than a “conventional” diet, a criterion that was permissible only for the control group. Exclusion criteria encompassed abstracts from society conferences, narrative reviews, systematic reviews, retrospective studies, animal studies, in vitro or in situ studies, and editorials. Additionally, studies were excluded if they lacked a solid or liquid food intervention, an appropriate control diet, or if they focused on baseline gastrointestinal symptoms attributed to conditions other than UC. Primarily, articles published in the English and Spanish languages were considered, spanning from the inception of relevant databases to the date of 1 July 2023. To discern the trajectory of knowledge advancement in this realm, antecedent studies referenced in the manuscript’s introduction were duly accounted for. Exclusion criteria were applied to studies involving patients diagnosed with Crohn’s Disease (CD) and/or Inflammatory Bowel Disease Unclassified (IBD-U), as well as investigations exclusively focusing on probiotics, nutritional supplementation, and parenteral nutrition. To enhance the search’s comprehensiveness, an additional step was executed, involving meticulous examination of the bibliographic references attached to prominent consensus documents and systematic reviews in the field.

We conducted a search with two independent reviewers in the following databases: PubMed, Embase, WOS, Cochrane Register of Controlled Trials (CENTRAL), CINHAL and Scopus. The search strategy used in PubMed was: “Colitis, Ulcerative” [Mesh] AND “Diet, Food, and Nutrition” [Mesh] AND “Therapeutics” [Mesh]. Adapted strategies were employed in other databases.

Two authors examined titles and abstracts and reached a consensus on the selection of articles. The selected articles were examined in full text.

### 2.3. Assessment of Risk of Bias and GRADE

The risk of bias in RCTs was independently assessed by two authors using the Cochrane Risk of Bias tool. The overall certainty of evidence for each stratified outcome was independently assessed by two authors using the GRADE methodology [22]. Inconsistencies were resolved by involving a third author.

### 2.4. Statistical Analysis

We utilized Review Manager 5.4 statistical software for conducting the meta-analysis. For continuous binary variables, the Risk Ratio (RR) or Odds Ratio (OR) along with their corresponding 95% confidence interval (CI) were employed in this study. If the index shows statistically significant variation across studies, the Random Effects Model (Random) was utilized for combining data. Conversely, if the heterogeneity test yielded a *p*-value greater than 0.05 and the *I*^2^ statistic was less than 50%, it indicated that there was no statistically significant heterogeneity across studies. In such cases, the Fixed Effects Model (Fixed) was employed for data integration.

## 3. Results

The systematic and comprehensive literature search identified an initial corpus of 3936 records, which underwent deduplication, resulting in 3208 unique records (Figure 1). After a rigorous screening of titles against predetermined criteria, 3084 records were excluded from the review. Subsequent abstract assessment led to the exclusion of 104 abstracts out of the initial 124, leaving 20 abstracts for full-text evaluation. Following a meticulous appraisal of these full-text articles, 14 studies were ultimately determined to meet the stringent inclusion criteria set forth in this systematic review [23,24,25,26,27,28,29,30,31,32,33,34,35,36]. Table 1 outlines the most important characteristics of the selected articles and Appendix A summarizes their risk of bias. In all the studies included, participants were allowed to continue their prescribed medication for inducing remission. This aspect distinguishes the dietary interventions analyzed in this review as adjunctive therapies—a distinct approach from that undertaken in CD. Both exclusive enteral nutrition and CD exclusion diet are designed to serve as therapeutic alternatives to pharmacological treatment in CD, presenting a divergent perspective [37].

### 3.1. Induction of Remission in Ulcerative Colitis

A total of 8 RCTs were included in the assessment the induction of clinical remission, involving a combined cohort of 288 patients. In 7 out of the 8 RCTs, no significant differences were observed between the interventions when compared to the standard diet (Figure 2).

Miyaguchi et al. [32] randomized 20 adolescent and adult patients with mild UC to receive either a non-restrictive diet, or a zinc enriched Japanese diet. The rates of remission at weeks 12 and 24 were notably higher in the group following the Japanese diet and zinc intake. Furthermore, the intervention group exhibited superior endoscopic improvement (as measured by the Ulcerative Colitis Endoscopic Index of Severity [UCEIS]) and histological improvement (measured by the Geboes Histological Score [GHS]). Sarbagilli-Shabat et al. [33], in a well-designed RCT, compared three groups of adult patients with active and refractory UC despite conventional treatments. They observed that clinical response rates, clinical remission rates, and endoscopic remission rates were higher in the group treated exclusively with Ulcerative Colitis Elimination Diet (UCED), although not reaching statistical significance. Candy et al. [25] reported higher rates of clinical remission in patients who underwent an elimination diet based on symptomatology. The exclusion of cow’s milk protein (RR, 1.22; 95% CI, 0.88–1.69; based on 2 RCTs comprising 106 participants) or gluten (RR, 1.42; 95% CI, 0.54–3.76; based on 1 RCT involving 51 participants) did not yield any statistically significant benefits in terms of inducing or maintaining a 52-week clinical remission [34,35]. Two RCT found a non-significant improvement in Partial Mayo Score (PMS) after 4–6 weeks of dietary treatment [26,30]. Regarding endoscopic remission, one RCT involving 77 participants, notable differences emerged, the exclusion of cow’s milk protein resulted in a more substantial 8-week endoscopic remission (RR, 2.02; 95% CI, 1.08–3.76), as did the exclusion of gluten (RR, 1.98; 95% CI: 1.03–3.79) but not UCED (RR, 2.27; 95% CI, 0.48–10.67) [24,26,36] (Appendix A). However, no notable difference was observed with regards to histologic remission in the context of the RCTs [25,29,32,36]. It’s crucial to highlight that the certainty of evidence for all these outcomes continued to be classified as very low, except for two RCTs that reached a moderate level of certainty [32,33] (Table 2).

### 3.2. Maintenance of Remission in Ulcerative Colitis

Five RCT [23,28,30,34,35], encompassing 110 participants with inactive UC, rigorously evaluated dietary interventions aimed at upholding clinical remission. No discernible advantages were observed from carrageenan-free diets, anti-inflammatory diets, or the elimination of cow milk protein for the maintenance of clinical remission over a 26- to 52-week period. Additionally, a positive outcome was observed in the study involving fermented cow milk with bifidobacterial [28]. However, it’s important to highlight that the pooled analysis did lean in favor of dietary intervention for sustaining clinical remission [RR, 0.7; CI 95%, 0.57–0.97] (Figure 3). The certainty of evidence for all these outcomes continued to be classified as low or very low (Table 3).

## 4. Discussion

From this comprehensive review, an intriguing conclusion emerges: the dietary interventions studied indeed demonstrate a favorable impact on the maintenance of clinical remission [23,28,30,34,35] [RR 0.75 (CI 95% 0.57–0.97), *I*^2^ = 24%], as well as positive outcomes in endoscopic [32,33,36] and histological remission [32,36]. However, the lack of a consistent correlation between these positive effects and clinical remission [RR 1.49 (CI 95% 0.96–2.31), *I*^2^ = 46%] underscores the intricate interplay of various factors, possibly including the differing assessment criteria employed in different studies.

The relationship between diet and UC emerges as a topic of compelling interest, fueled by the increasing recognition of how specific dietary components can influence gut health and microbial balance. However, while the connection between diet and gut health is becoming clearer, the precise role of diet in managing UC remains a matter of ongoing investigation [1].

Contemporary insights into the gut-microbiota axis have illuminated the intricate interplay between dietary factors and intestinal health. While certain dietary constituents have been implicated in exerting detrimental effects on gut homeostasis, the translation of this knowledge into tailored dietary strategies for managing UC remains far from definitive. Our comprehensive analysis underscores the need to understand how diet complements conventional therapeutic approaches.

A central issue that emerges from our review is the variability in the methodological rigor of the included RCTs. The heterogeneity in study design, patient populations, and outcome measures complicates the task of deriving coherent conclusions from the collective body of evidence. This limitation, while acknowledging the insights gleaned from individual studies, underscores the necessity for methodologically robust research that can withstand scrutiny and enhance the precision of recommendations in this clinical domain [39].

Furthermore, contextual issues should be considered when interpreting the findings of these studies. Notably, all examined studies employed dietary interventions alongside other pharmacological treatments to induce remission, such as corticosteroids or 5-aminosalicylic acid (5-ASA). This concomitant treatment approach introduces an inherent challenge in isolating the independent effects of diet. As evidenced by the effectiveness of the complete diet elimination therapy, Crohn’s Disease Exclusion Diet (CDED) in CD, untangling these complex interactions is vital for refining dietary interventions in UC management [40].

Temporal considerations also warrant attention. A significant subset of studies included in this review spans nearly six decades, reflecting the evolving diagnostic criteria and standards for clinical remission [32,35]. The disparity between historical and contemporary criteria introduces a potential confounder, necessitating cautious interpretation of the findings and highlighting the importance of aligning research methodologies with current clinical paradigms.

Considering the evolving evidence, crafting concrete dietary recommendations for UC management remains a formidable challenge. The consensus document from the International Organization for the Study of Inflammatory Bowel Diseases (IOIBD) suggests several dietary recommendations aimed at managing UC [1]. These recommendations are based on the growing understanding of how specific dietary choices can impact the disease and overall gut health. One key recommendation is to increase the consumption of omega-3 fatty acids, found in sources like fish oil and fresh fish. Omega-3 fatty acids are known for their anti-inflammatory properties and have been shown to potentially alleviate symptoms and promote better gut health in individuals with UC [1]. Conversely, the consensus advises reducing the consumption of certain foods and substances that could exacerbate the condition. These include red meats and processed meats, which are associated with increased inflammation. The recommendation to limit dairy fat aligns with the idea of reducing saturated fat intake, as saturated fats are linked to inflammation and could potentially worsen symptoms [1]. The consensus also advocates for cutting back on certain types of fats, such as those found in coconut oil and palm oil, as well as saturated fats and trans fats. These fats are often found in processed and fried foods and have been associated with promoting inflammation and gut distress. Furthermore, the document recommends avoiding emulsifiers, carrageenan, artificial sweeteners, maltodextrins, and titanium dioxide. These additives, commonly found in processed foods, have been linked to potential disruption of gut barrier function and gut microbiota balance, which could contribute to disease progression or symptoms in individuals with UC [1]. The Ulcerative Colitis Exclusion Diet (UCED) augments recommendations for UC management. It advises decreasing sulphated amino acids, animal protein, haeme, animal and saturated fats, and food additives. Instead, increasing tryptophan intake, along with natural sources of pectin and resistant starch, is suggested. These modifications align with anti-inflammatory principles and gut health promotion [41]. In recent studies [33,41], clinical outcomes following UCED treatment were consistent across both adults and children: 60% of adults achieved a clinical response compared to 71% of children, while 40% of adults and 38% of children achieved clinical remission. Subsequently, it was deemed that combining UCED with personalized enteral nutrition (PEN) would offer complementary nutritional support and enhance outcomes by aiding dietary intake, balancing micro and macronutrients, promoting adherence and efficacy, and optimizing fiber consumption. In this scenario, a current multicenter randomized controlled trial (ReDUCE), registered on ClinicalTrials.gov (ID: NCT05791487), seeks to determine whether the combination of UCED and PEN can enhance outcomes when administered alongside an oral budesonide regimen for adults with mild to moderate UC.

Integrating UCED suggestions with IOIBD consensus guidelines provides a comprehensive dietary strategy. Consulting a specialized dietitian for personalized adjustments is vital. Adhering to these recommendations empowers individuals to make informed dietary choices that may alleviate symptoms and enhance their overall well-being while managing UC [33].

This systematic review exhibits several strengths, including a rigorous methodology for systematic review and meta-analysis, a comprehensive evaluation of relevant dietary trials, and meticulous data analysis and interpretation. However, several limitations must be acknowledged. Firstly, despite efforts to address it, clinical and methodological heterogeneity persisted among the studies, which could not be eliminated in the pooled analyses. Stratification based on dietary principles and reporting of statistical heterogeneity were employed, but caution is warranted when interpreting pooled data due to the existing heterogeneity. Secondly, the influence of medications and unmeasured confounders on the analyzed outcomes remained a potential concern, with partial mitigation in randomized controlled trials but not in observational studies. Thirdly, missing data due to incomplete collection or reporting by some studies hindered quantitative analysis for all included studies.

## 5. Conclusions

The relationship between diet and UC is an ever-evolving area of study that demands deeper exploration. While we are accumulating valuable insights into the potential influence of diet on this disease, a complete and conclusive understanding has not yet been achieved, due to inconclusive published data. The synthesis of evidence from this systematic review highlights the need for rigorous research that bridges the gap between theory and practice. Until diet supplants medical treatment, adopting a thoughtful and individualized dietary approach could contribute to symptom alleviation, and the promotion of a more favorable disease course.

## Figures and Tables

**Figure 1 nutrients-15-04194-f001:**
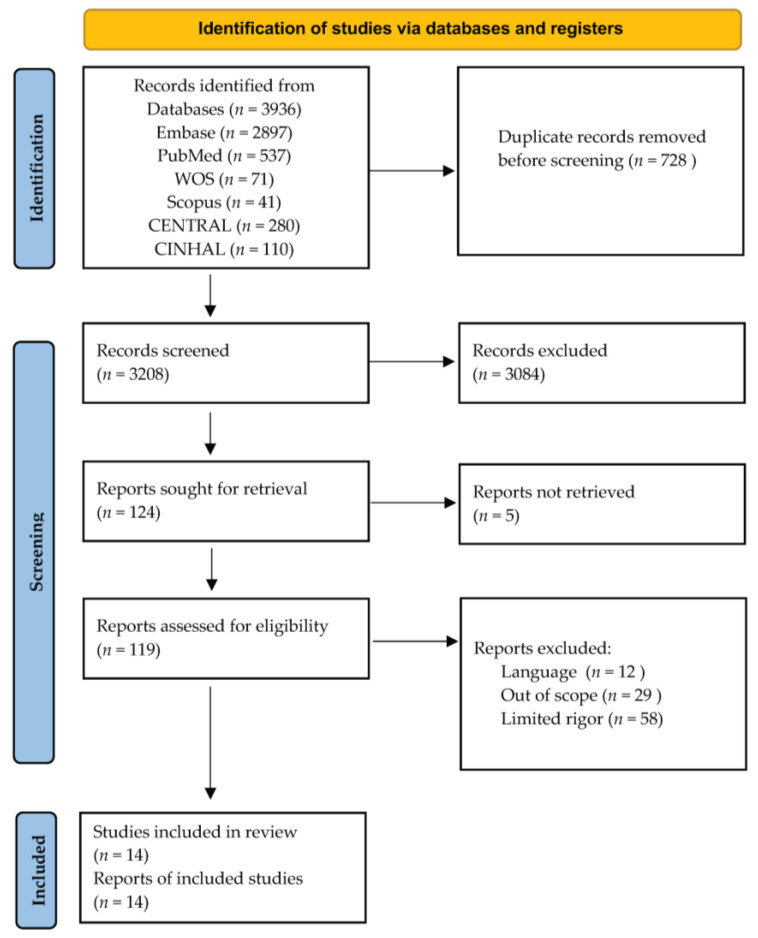
Selection process of records reporting the effect of the diet in the treatment of ulcerative colitis. The flowchart was based on the Preferred Reporting Items for Systematic Reviews and Meta-Analyses (PRISMA) guidelines [20]. Automation tools were not used.

**Figure 2 nutrients-15-04194-f002:**
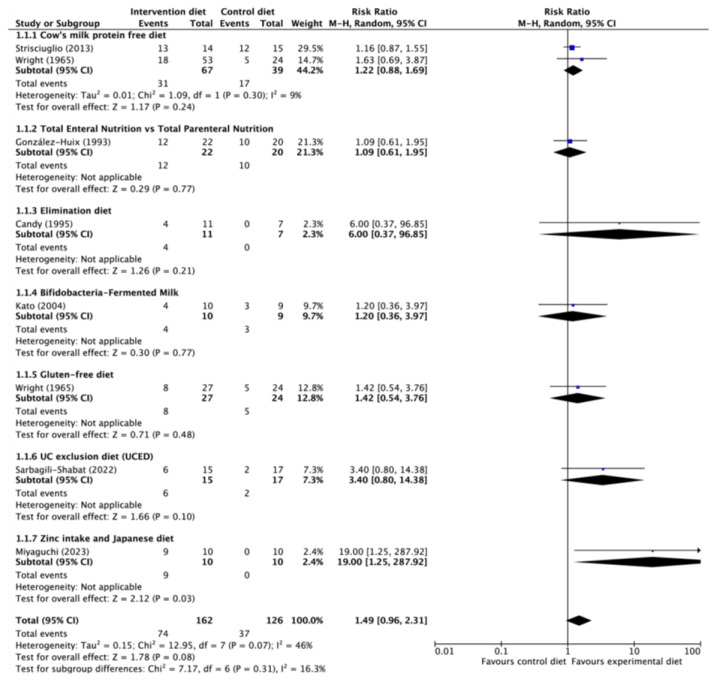
Induction of clinical remission in ulcerative colitis [25,27,29,32,33,34,35].

**Figure 3 nutrients-15-04194-f003:**
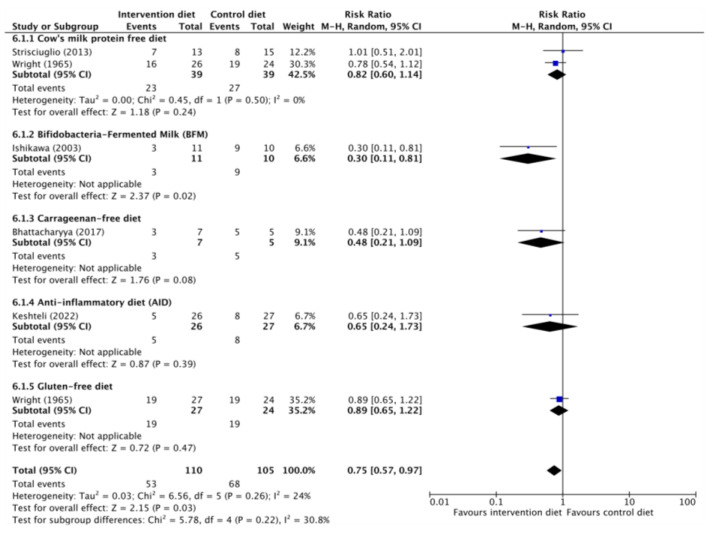
Clinical relapses in ulcerative colitis [23,28,30,34,35].

**Table 1 nutrients-15-04194-t001:** Characteristics of selected articles.

First Author (Year Publication) [Ref.]	Study Population	Intervention	Duration	Variables	Outcomes	Conclusions
Wright (1965) [35]	77 active UC adult patients (26 milk-free diet, 24 dummy diet and 27 gluten free diet). GFD was not milk-free diet.	Milk free, low-roughage diet vs. dummy diet or gluten-free and milk-free diet ^(1)^	12 months	Clinical course (number of relapses during the trial period). Clinical remission	Milk free diet vs. dummy diet Clinical remission: 18/53 (33.9%) vs. (20.8%). [RR = 1.63 (CI 95% 0.69–3.87)] No relapses: 10/26 (38.4%) vs. 5/24 (20.8%). [RR = 1.84 (IC 95% 0.73–4.62)] Gluten free diet vs. dummy diet Clinical remission: 8/27 (29.6%) vs. 5/24 (20.8%). RR = 1.42 (CI 95% 0.54–3.76)] No relapses: 8/27 (29.6%) vs. 5/24 (20.8%). [RR = 1.42 (IC 95% 0.54–3.76)]	The milk-free diet and the gluten-free diet were superior to dummy diet to avoid relapses in patients with UC
Wright (1966) [36]	77 active UC adult patients (26 milk-free diet, 24 dummy diet and 27 gluten free-diet). GFD was not milk-free diet.	Milk free diet vs. Milk Milk free, low-roughage diet vs. dummy diet or gluten-free and milk-free diet ^(1)^	12 months	Sigmoidoscopic remission, histologic remission	Milk free diet vs. dummy diet Endoscopic remission: 18/25 (72%) vs. 7/20 (35%). [RR = 2.05 (CI 95% 1.07–3.92)] Histologic remission: 9/17 (52.9%) vs. 3/14 (21.4%). [RR = 2.47 (CI 95% 0.82–7.41)] Gluten free diet vs. dummy diet Endoscopic remission: 18/26 (69.2%) vs. 7/20 (35%). [RR = 1.98 (CI 95% 1.03–3.79)] Histologic remission: 7/18 (38.8%) vs. 3/14 (21.4%). [RR = 1.81 (CI 95% 0.57–5.78)]	The milk-free and gluten-free diet were superior to dummy diet in inducing endoscopic and histologic remission
González-Huix (1993) [27]	42 active UC adult patients (22 steroids and TEN vs. 20 steroids and TPN)	TEN vs. TPN	Until remission or colectomy	Clinical remission (Truelove’s index)	Remission rates: 54.5% vs. 50.0%. *p* = 0.764. [RR = 1.09 (IC 95% 0.61–1.95)] More complications after colectomy and related with artificial nutritional support in TPN group	TEN is safe and effective in patients with severe UC. Compared with TPN, is cheaper and is associated with a lower complication rate.
Candy (1995) [25]	18 active UC adult patients (11 intervention and 7 in control group).	Elimination diet ^(2)^	6 weeks	Remission and improvement rates. Sigmoidoscopy and biopsy findings.	Remission rates: 36.6% vs. 0%, *p* = 0.0009 Improvement rates: 45.4% vs. 14.28% [RR = 3.2 (CI 95% 0.46–21.84)]. No improvement: 18.2% vs. 85.7% [RR = 1.27 (CI 95% 0.14–11.5)]. Remission or improvement rates: 84.6% vs. 14.2%. [RR = 5.9 (CI 95% 0.9–36.8)]. Endoscopic improvement: 72.2% vs. 33.3%. [RR = 2.1 (CI 95% 0.6–7.1)]. Histologic improvement: 27.2% vs. 50%. [RR = 0.5 (CI 95% 0.1–1.9)].	Patients with mild to moderate UC may be brought into remission by the manipulation of dietary intake.
Ishikawa (2003) [28]	21 UC adult patients in remission BFM GROUP: 11 CONTROL: 10	Bifidobacterium-Fermented Milk (BFM, 100 mL per day) + conventional treatment vs. conventional treatment	12 months	Relapse during the follow-up	Exacerbations of symptoms was seen in 27.3% of subjects in the BFM vs. 90% of control group. Cumulative exacerbation rates were higher in the control group RR: 3.3 (CI 95%, 1.23–8.85), *p* = 0.0075. Log rank = 0.0184	Supplementation with the BFM product was successful in maintaining remission and had possible preventive effects on the relapse of ulcerative colitis.
Kato (2004) [29]	20 active UC adult patients: BFM GROUP: 10 CONTROL: 10	Bifidobacterium-Fermented Milk (BFM, 100 mL per day) + conventional treatment vs. conventional treatment	12 weeks	Clinical response Clinical remission Endoscopic outcomes Histologic outcomes	Response rates: 70% vs. 33% [RR = 2.1 (IC 95% 1.54–2.85)] Remission rates: 40% vs. 33% [RR = 1.33 (IC 95% 0.39–4.48)] The CAI score in the BFM group was significantly lower than that of the placebo group at 12 weeks (*p* < 0.05). The endoscopic activity index score in the BFM group was not significantly different from that of the placebo group at 12 weeks. The histological score in the BFM group was also significantly reduced from 4.4 ± 0.3 to 3.1 ± 0.3 after 12 weeks of treatment (*p* < 0.01) There was no significant difference between the values before vs. 12 weeks after starting treatment in the placebo group.	Supplementation with this bifidobacteria-fermented milk product is safe and more effective than conventional treatment alone, suggesting possible beneficial effects in managing active ulcerative colitis.
Strisciuglio (2013) [34]	29 pediatric patients newly diagnosis of UC. 14 received CMP elimination diet and 15 free diet.	Cow’s milk protein-free diet vs. usual diet plus conventional treatment: If PUCAI < 35: 5-ASA treatment If PUCAI ≥ 35: steroids and 5-ASA	12 months	Clinical remission (PUCAI)	Remission rates: 92.8% vs. 80% [RR = 1.16 (CI 95% 0.86–1.55)] Relapse rates within 1 year of follow-up: 53.8% vs. 53.3% [RR = 0.99 (CI 95% 0.49–1.97)]	The elimination of cow’s milk proteins from the diet has no relevant role in the induction and maintenance of remission of UC in children.
Kyaw (2014) [31]	112 active UC adult patients	Specific dietary Guidelines for UC (61 patients) vs. usual diet (51 patients)	4–6 weeks	Disease activity (SCCAI), HR-QOL (IBDQ)	Significant reduction in SCCAI in the intervention group (*p* = 0.0108) compared to control. No significant changes in quality of life	Probable association between specific dietary advice for UC and symptom improvement
Bhattacharyya (2017) [23]	12 adult UC patients in clinical remission	Carrageenan-free diet vs. Carrageenan-free diet with capsules containing 200 mg food-grade carrageenan	12 months	Clinical relapse (SCCAI), calprotectin, SIBDQ	Fewer relapses in the placebo group (0% vs. 60%, *p* = 0.046). Increased levels of IL-6 (*p* = 0.02) and faecal calprotectin (*p* = 0.06) in the carrageenan group. No differences in quality of life.	Carrageenan intake contributed to higher relapse frequency in UC patients in remission
Bodini (2019) [24]	20 adult UC patients (clinical remission or mild disease)	Low FODMAPsdiet	6 weeks	Clinical remission (PMS), calprotectin, HRQOL(IBD-Q)	At baseline 14 (70%) patients were in remission and 6 (30%) had mild disease. After 6 weeks 17 (85%) were in clinical remission and 3 (15%) had mild disease. No significant differences in PMS from baseline to 6 weeks in both groups. After the 6-wk dietary intervention, a statistically significant decrease in median calprotectin values in the LFD group was observed (*p* = 0.004). After the 6-wk dietary intervention, a modest but statistically significant increase in median IBD-Q in the LFD group was observed.	A short-term low FODMAP diet (LFD) is safe for UC patients and is linked to improved fecal inflammatory markers and quality of life, even in those with mainly quiescent disease. It can also induce clinical remission in patients with mild disease.
Fritsch (2021) [26]	18 adult patients with inactive or mild UC	Low fiber, high-fiber diet (LFD) vs. An improved standard American Diet (iSAD)	10 weeks 2 weeks of washout between 2 periods of 4 weeks	Clinical remission (PMS), CRP, calprotectin, HR-QOL (SIBDQ)	Significant improvement in quality of life according to SIBDQ in the LFD diet (*p* = 0.02) and iSAD (*p* = 0.001). Significant improvement in SF-36 with both diets. Decrease in CRP (*p* = 0.07) and serum amyloid A protein (*p* = 0.02) in LFD. After LFD, the relative abundance of Actinobacteria decreased (*p* = 0.017), while that of Bacteroidetes increased (*p* = 0.015).	Both diets are well tolerated with improvements in quality of life. The low-fat, high-fibre diet decreased markers of inflammation, gut dysbiosis and PMS.
Keshteli (2022) [30]	53 adults with UC in CR (PMS ≤ 2 points with a rectal bleeding subscore ≤ 1) in the previous 18 months	Anti-inflammatory diet (AID) *n* = 26 vs. Canada’s Food Guide (CFG) *n* = 27	6 months	Clinical relapse (PMS), Subclinical response (Fcal < 150 μg/g) HR-QOL (SIBDQ)	Clinical relapse: 19.2% AID vs. 29.6% (CFG), *p* = 0.38. [RR = 0.649 (CI 95% 0.24–1.72)] The SIBDQ scores (to assess quality of life) did not change significantly from the baseline to the last visit, either in the control group (5.5 ± 0.7 vs. 5.5 ± 0.9, *p* = 0.80) or in the AID group (5.5 ± 0.9 vs. 5.6 ± 0.8, *p* = 0.56) The subclinical response was significantly higher in the AID group in comparison to the CFG group (69.2 vs. 37.0%, *p* = 0.02). [RR = 1.86 (CI 95% 1.07–3.25)]	The AID was effective in preventing subclinical inflammation.
Sarbagili-Shabat (2022) [33]	Adults with active and refractory established UC. Arm 1 (*n* = 17) Arm 2 (*n* = 19) Arm 3 (*n* = 15)	Arm 1: free diet and standard FT without dietary conditioning of the donor. Arm 2: FT with dietary pre-conditioning of the donor for 14 days and UCED Arm 3: UCED without FT	12 weeks	Clinical remission (SCCAI), Clinical improvement, endoscopic remission.	Clinical response (arms 1,2,3): 35.3% vs. 42.1% vs. 60% [RR_(3vs2)_ = 1.44 (CI 95% 0.69–2.99)] [RR_(3vs1)_ = 1.61 (CI 95% 0.79–3.29)] Clinical remission (arms 1,2,3): 11.8% vs. 21.1% vs. 40% [RR_(3vs2)_ = 1.9 (CI 95% 0.65–5.53)] [RR_(3vs1)_ = 3.4 (CI 95% 0.84–14.3)] Endoscopic remission (arms 1,2,3): 11.8% vs. 15.8% vs. 26.7% [RR_(3vs2)_ = 1.68 (CI 95% 0.44–6.41)] [RR_(3vs1)_ = 2.26 (CI 95% 0.48–10.66)]	UCED alone appeared to achieve higher clinical remission and mucosal healing than single donor FT with or without diet.
Miyaguchi (2023) [32]	Teenagers and adults with mild UC	Unrestricted diet (UD) vs. promotion of zinc intake and Japanese diet (JD)	24 weeks	Clinical (CAI), Endoscopic (UCEIS) and histological (GHS) remissions	Clinical remission at 12w: 20% vs. 0% Clinical remission at 24w: 90% vs. 8% UCEIS score: JD: T0: 2.5 ± 0.7; T12w: 1.7± 0.7; T24w: 1.6 ± 0.7, *p* = 0.021; *p* = 0.02 UD: T0:2.1 ± 0.6; T12w: 2.0 ± 0.5; T24w: 1.9 ± 0.6, *p* = ns; *p* = ns GHS: JD: T0: 2.6 ± 0.5; T12w: 2.0 ± 0.8; T24w:1.0 ± 0.7, *p* = 0.048; *p* = 0.0008 UD: T0: 2.4 ± 0.5; T12w: 2.2 ± 0.4; T24w: 1.6 ± 1.0, *p* = 0.894; *p* = 0.071	Promotion of zinc intake and a Japanese diet rich in *n*-3 fatty acids may induce clinical remission in patients with mild active UC

^(1)^ In the gluten and cow’s milk-free diet, traces of cow milk were detected, thus rendering it ineligible for classification as cow’s milk-free diet. ^(2)^ Elimination diet: In the initial week of the study, all dairy products were removed from the participants’ diet, only to be reintroduced gradually thereafter. Throughout the study period, the consumption of refined sugars, preservatives, additives, spices and condiments was strictly prohibited, with the exception of salt. Additionally, only boiled water was permitted as a beverage option. Any foods that triggered symptoms in the participants were immediately eliminated from their diet. The prescribed diet primarily consisted of a carefully curated selection of fruits, vegetables, grains, meats, and fish, which could be prepared using any method except frying. GFD: gluten-free diet; CAI: clinical activity index; UC: ulcerative colitis; FODMAP: fermentable oligosaccharides, disaccharides, monosaccharides and polyols; HR: hazard ratio; IBDQ: inflammatory bowel disease quality of life questionnaire; CRP: *C*-reactive protein; PUCAI: pediatric ulcerative colitis activity index; SCCAI: simple clinical activity index for ulcerative colitis; SF-36: short quality of life questionnaire 36; SIBDQ: short inflammatory bowel disease quality of life questionnaire; UCDAI: ulcerative colitis disease activity index; PMS: Partial Mayo Score; UCDED: Ulcerative Colitis Exclusion Diet.

**Table 2 nutrients-15-04194-t002:** Summary of findings: Induction of Clinical Remission in Ulcerative Colitis.

Certainty Assessment							№ of Patients		Effect		Certainty	Importance
№ of Studies	Study Design [Ref.]	Risk of Bias	Inconsistency	Indirectness	Imprecision	Other Considerations	Intervention Diet	Control Diet	Relative (95% CI)	Absolute (95% CI)		
DIET: Elimination diet												
1	RCT [19]	serious ^a^	not serious	not serious	very serious ^b^	none	7/11 (63.6%)	0/7 (0.0%)	RR 1.66 (0.85 to 3.23)	0 fewer per 1000 (from 0 fewer to 0 fewer)	⨁◯◯◯ Very low	IMPORTANT
DIET: Cow’s milk protein free diet												
2	RCT [28,29]	serious ^a^	not serious	not serious	very serious ^b^	none	31/67 (46.3%)	0.0%	RR 1.16 (0.87 to 1.55)	0 fewer per 1000 (from 0 fewer to 0 fewer)	⨁◯◯◯ Very low	IMPORTANT
DIET: Gluten-free diet												
1	RCT [29]	serious ^a^	not serious	not serious	very serious ^b^	none	8/27 (29.6%)	5/24 (20.8%)	RR 1.09 (0.61 to 1.95)	19 more per 1000 (from 81 fewer to 198 more)	⨁◯◯◯ Very low	IMPORTANT
DIET: Total Enteral Nutrition vs. Total Parenteral Nutrition												
1	RCT [21]	very serious ^c^	serious ^d^	not serious	very serious ^b^	none	12/22 (54.5%)	10/20 (50.0%)	RR 6.00 (0.37 to 96.85)	1000 more per 1000 (from 315 fewer to 1000 more)	⨁◯◯◯ Very low	IMPORTANT
DIET: Bifidobacteria-Fermented Milk												
1	RCT [23]	very serious ^c^	not serious	not serious	very serious ^d^	none	4/10 (40.0%)	3/9 (33.3%)	RR 1.20 (0.36 to 3.97)	67 more per 1000 (from 213 fewer to 990 more)	⨁◯◯◯ Very low	IMPORTANT
DIET: UC exclusion diet (UCED)												
1	RCT [27]	not serious	not serious	not serious	serious ^e^	none	6/15 (40.0%)	2/17 (11.8%)	RR 3.40 (0.80 to 14.38)	282 more per 1000 (from 24 fewer to 1000 more)	⨁⨁⨁◯ Moderate	IMPORTANT
DIET: Zinc intake and Japanese diet vs. unrestricted diet												
1	RCT [26]	not serious	not serious	not serious	serious ^e^	none	9/10 (90.0%)	0/10 (0.0%)	RR 19.00 (1.25 to 287.92)	0 fewer per 1000 (from 0 fewer to 0 fewer)	⨁⨁⨁◯ Moderate	IMPORTANT
DIET: Low fat, low carbohydrate, low fiber, high protein and probiotics (DMF)												
1	RCT [32]	not serious	serious ^d^	not serious	very serious ^b^	none	61	51	-	mean 1.304 points from baseline more (0.21 more to 2.398 more)	⨁◯◯◯ Very low	IMPORTANT
DIET: Low FODMAPs												
1	RCT [18]	not serious	serious ^d^	not serious	very serious ^b^	none	PMS decreased after the 6-wk dietary intervention, although not significantly, in the Low FODMAPs Diet group (*n* = 8 patients vs. 12 patients in the control group).				⨁◯◯◯ Very low	IMPORTANT
DIET: Low fat, high fiber												
1	RCT [33]	not serious	serious ^d^	not serious	very serious ^b^	none	PMS decreased after the 4-wk dietary intervention although not significantly, in the low fat and high fiber diet group (*n* = 18 patients)				⨁◯◯◯ Very low	IMPORTANT

**CI:** confidence interval; **RR:** risk ratio; **RCT:** randomized control trial; **PMS:** Partial Mayo Score. Importance and certainty of evidence: The GRADE approach suggests rating the importance of each outcome on a 9-point scale from 1 to 9. A rating or assessment of the importance of outcomes is necessary to choose which outcomes should be considered in deciding about the benefits and downsides of an intervention or about which outcomes should be included in a GRADE evidence profile or Summary of Findings table. The certainty of evidence reflects the extent to which our confidence in an estimate of the effect is adequate to support a particular recommendation [38]. Explanations: ^a^. Downgraded 1 level due to serious limitations; ^b^. Downgraded 2 levels due to very serious imprecision; ^c^. Downgraded 2 levels due to high or unclear risk of bias; ^d^. Downgraded 1 level due to serious inconsistence; ^e^. Downgraded 1 level due to serious imprecision.

**Table 3 nutrients-15-04194-t003:** Summary of Findings: Clinical relapse in Ulcerative Colitis.

Certainty Assessment							№ of Patients		Effect		Certainty	Importance
№ of Studies	Study Design	Risk of Bias	Inconsistency	Indirectness	Imprecision	Other Considerations	Clinical Relapse in Ulcerative Colitis	Placebo	Relative (95% CI)	Absolute (95% CI)		
DIET: Cow’s milk protein free diet												
2	RCT [28,29]	serious ^a^	not serious	not serious	very serious ^b^	none	23/39 (59.0%)	27/39 (69.2%)	RR 0.82 (0.60 to 1.14)	125 fewer per 1000 (from 277 fewer to 97 more)	⨁◯◯◯ Very low	IMPORTANT
DIET: Bifidobacteria-Fermented Milk (BFM)												
1	RCT [22]	very serious ^c^	not serious	not serious	very serious ^b^	none	3/11 (27.3%)	9/10 (90.0%)	RR 0.30 (0.11 to 0.81)	630 fewer per 1000 (from 801 fewer to 171 fewer)	⨁◯◯◯ Very low	IMPORTANT
DIET: Carrageenan-free diet												
1	RCT [17]	very serious ^c^	very serious ^b^	not serious	serious ^d^	none	3/7 (42.9%)	5/5 (100.0%)	RR 0.48 (0.21 to 1.09)	520 fewer per 1000 (from 790 fewer to 90 more)	⨁◯◯◯ Very low	IMPORTANT
DIET: Anti-inflammatory diet (AID)												
1	RCT [24]	serious ^a^	not serious	not serious	serious ^d^	none	5/26 (19.2%)	8/27 (29.6%)	RR 0.65 (0.24 to 1.73)	104 fewer per 1000 (from 225 fewer to 216 more)	⨁⨁◯◯ Low	IMPORTANT

**CI:** confidence interval; **RR:** risk ratio; **RCT**: randomized control trial. Importance and certainty of evidence: The GRADE approach suggests rating the importance of each outcome on a 9-point scale from 1 to 9. A rating or assessment of the importance of outcomes is necessary to choose which outcomes should be considered in deciding about the benefits and downsides of an intervention or about which outcomes should be included in a GRADE evidence profile or Summary of Findings table. The certainty of evidence reflects the extent to which our confidence in an estimate of the effect is adequate to support a particular recommendation [38]. Explanations: ^a^. Downgraded 1 level due to serious risk of bias. ^b^. Downgraded 2 levels due to very serious imprecision. ^c^. Downgraded 2 levels due to high or unclear risk of bias. ^d^. Downgraded 1 level due to serious imprecision.

## Data Availability

Not applicable.

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
