# Peer review of "Dietary Interventions in Ulcerative Colitis: A Systematic Review of the Evidence with Meta-Analysis"

_nutrients, 2023, doi:10.3390/nu15194194_

Round 1
Reviewer 1 Report
This comprehensive systematic review and meta-analysis focused on the impact of dietary interventions in the management of ulcerative colitis (UC). By analyzing a diverse range of studies spanning several decades, the study aimed to shed light on the intricate relationship between diet and UC outcomes. The findings revealed that dietary interventions, such as the Ulcerative Colitis Exclusion Diet (UCED), have shown promise in inducing clinical and endoscopic remission, particularly in cases of milder refractory disease. However, the evidence also highlighted the complexity of UC and the variability in patient responses to dietary changes. While dietary strategies, such as increasing omega-3 fatty acids and reducing inflammatory foods, have shown potential benefits, the study emphasized the need for further research, methodological rigor, and personalized dietary recommendations to optimize UC management. This study serves as a valuable contribution to the ongoing exploration of dietary approaches in UC care, emphasizing the potential for tailored dietary strategies to complement conventional treatments and enhance overall patient well-being.
General Comments:
Title and Abstract: The title of the paper is clear and informative, providing a concise overview of the research topic. The abstract provides a good summary of the study's objectives and key findings.
Introduction: The introduction sets the stage well by highlighting the importance of the topic and the existing knowledge gap. However, it could be further improved by providing a more specific research question or hypothesis.
Methodology: The description of the methodology, including the inclusion criteria for studies and the search strategy, is comprehensive and clear. However, it might benefit from a brief explanation of the rationale behind the inclusion and exclusion criteria.
Results: The presentation of results is clear and well-organized. Tables and figures are appropriately used to illustrate key findings. However, some sections of the results could be condensed for conciseness.
Discussion: The discussion section provides a thorough analysis of the findings, including their implications and limitations. It effectively addresses the complexity of the topic and the need for further research. However, it might benefit from a more explicit connection between the findings and their practical implications for clinicians and patients.
Conclusion: The conclusion summarizes the key takeaways from the study. It might be enhanced by briefly restating the main findings and their significance.
Specific Comments:
Clarity and Language: The paper is generally well-written and easy to understand. However, there are some lengthy sentences and complex language that could be simplified for greater clarity.
References: The paper could benefit from a more extensive discussion of the relevant literature, particularly recent studies or reviews related to dietary interventions in ulcerative colitis.
Recommendations: Consider including practical recommendations or implications for clinical practice, such as dietary guidelines for patients with ulcerative colitis based on the findings.
Future Research: While the discussion mentions the need for further research, consider specifying potential directions for future studies or hypotheses that arise from the current findings.
“The APC was funded by Nestlé Health Science”. it's important to ensure that this funding source does not influence the research or its findings in any way that would compromise the integrity of the study. Research should be conducted independently and objectively, regardless of the funding source.

English is good
Author Response
REVIEWER 1
AUTHORS: We would like to thank to Reviewer 1 the effort and time devoted to our manuscript. Thank you for the constructive comments that obviously have improved our manuscript.
General comments
- Title and Abstract: The title of the paper is clear and informative, providing a concise overview of the research topic. The abstract provides a good summary of the study's objectives and key findings. Thank you.
- Introduction: The introduction sets the stage well by highlighting the importance of the topic and the existing knowledge gap. However, it could be further improved by providing a more specific research question or hypothesis. Thank you for your comment. We have added this text at the end of the background section: This comprehensive systematic review endeavors to scrutinize and synthesize the existing evidence regarding the impact and efficacy of various diets in the management and treatment of UC in both pediatric and adult populations. The research question is set out in the material and methods section.
- Methodology: The description of the methodology, including the inclusion criteria for studies and the search strategy, is comprehensive and clear. However, it might benefit from a brief explanation of the rationale behind the inclusion and exclusion criteria. Thank you for your comment. We have added this paragraph to the text: Studies selected for review were confined to RCTs investigating the effects of defined solid or liquid diets, as compared to a control diet, in individuals with UC. The intervention group was mandated to adhere to a rigorously delineated diet, rather than a "conventional" diet, a criterion that was permissible only for the control group. Exclusion criteria encompassed abstracts from society conferences, narrative reviews, systematic reviews, retrospective studies, animal studies, in vitro or in situ studies, and editorials. Additionally, studies were excluded if they lacked a solid or liquid food intervention, an appropriate control diet, or if they focused on baseline gastrointestinal symptoms attributed to conditions other than UC
- Results: The presentation of results is clear and well-organized. Tables and figures are appropriately used to illustrate key findings. However, some sections of the results could be condensed for conciseness. Thank you for the comment. We have shortened the results.
- Discussion: The discussion section provides a thorough analysis of the findings, including their implications and limitations. It effectively addresses the complexity of the topic and the need for further research. However, it might benefit from a more explicit connection between the findings and their practical implications for clinicians and patients. I completely agree with you. The discussion has been shortened and there has been an effort to focus the topic and correlate it with practical points.
- Conclusion: The conclusion summarizes the key takeaways from the study. It might be enhanced by briefly restating the main findings and their significance. DONE
Specific Comments
Comment
Clarity and Language: The paper is generally well-written and easy to understand. However, there are some lengthy sentences and complex language that could be simplified for greater clarity. Thank you so much. DONE.
Comment
References: The paper could benefit from a more extensive discussion of the relevant literature, particularly recent studies or reviews related to dietary interventions in ulcerative colitis. I’m totally agree with your suggestion and will take it into consideration for the narrative review we are conducting. In this article, due to the methodology used and the limited publications available, we believe we cannot make more recommendations than those we attribute and reference in the discussion. Thank you.
Comment
Recommendations: Consider including practical recommendations or implications for clinical practice, such as dietary guidelines for patients with ulcerative colitis based on the findings. Thank you very much for your comment. We are working on another article, a narrative review, where we will include these aspects. Thank you.
Comment
Future Research: While the discussion mentions the need for further research, consider specifying potential directions for future studies or hypotheses that arise from the current findings. DONE. Thank you for your comment.
Comment
“The APC was funded by Nestlé Health Science”. it's important to ensure that this funding source does not influence the research or its findings in any way that would compromise the integrity of the study. Research should be conducted independently and objectively, regardless of the funding source. Thank you for your comment. We have added this paragraph: “The authors want to emphasize that the source of funding has had no impact on the research or its conclusions. The research has been carried out independently and objectively, completely unaffected by the funding source.”
Reviewer 2 Report
This is a systematic review with clinical relevance. The methodology is appropriate; however, the discussion needs further exploration. It would be beneficial to delve into the specific characteristics of the diets included in the meta-analysis rather than attempting to discuss the overall outcome. This approach could be expanded for each nutritional therapy evaluated.
Therefore, I suggest some general changes:
Throughout the text, the authors discuss dietary treatment on an individual basis to reach general conclusions. This is not appropriate, since the meta-analyses involved the different dietary alternatives used and were analyzed as a single unit (dietary therapy). Therefore, reaching conclusions with specific dietary treatments is beyond the methodological possibilities of the article.
Suggestions to authors:
*Remove "of the evidence (GRADE)" from the title;
*Add the number of included studies to the summary
*In the introduction, include in the direct effects of food the pro-oxidant effects and in connection with inflammatory action;
*Even in the introduction, exclude the description of specific objectives;
*Please use the "Flow diagram of study selection Prism" template;
*delete column "Study design" as being RCT is an inclusion criterion;
*As having UC is an inclusion criterion, there is no need to describe this fact in the results table, but it is essential to include the mean and SD of age;
*What is "Elimination diet" by the author Candy (1995);
*In line 146, delete the term "Randomized Controlled Trials" and leave only the acronym RCT

Author Response
AUTHORS: We would like to thank to Reviewer 2 the effort and time devoted to our manuscript. Thank you for the constructive comments that obviously have improved our manuscript.
Comment
Throughout the text, the authors discuss dietary treatment on an individual basis to reach general conclusions. This is not appropriate, since the meta-analyses involved the different dietary alternatives used and were analyzed as a single unit (dietary therapy). Therefore, reaching conclusions with specific dietary treatments is beyond the methodological possibilities of the article. Dear reviewer, we appreciate your concern regarding the methodology and the approach to individual dietary treatments. The intention behind examining dietary treatments on an individual basis was to provide an initial exploration into the potential effects of each specific diet. We understand that the meta-analyses grouped these diets together under the broader category of "dietary therapy." However, in our opinion, this approach represents an initial and valid attempt to evaluate the effect of different diets on UC.
Suggestions to authors:
- *Remove "of the evidence (GRADE)" from the title;
- *Add the number of included studies to the summary. We have added this sentence: Fourteen RCTs were included
- *In the introduction, include in the direct effects of food the pro-oxidant effects and in connection with inflammatory action; Thank you for this point. We have added this to the text: “Long-term intake of high-fat or high-sugar diets can lead to the production of excessive free radicals. This initiates a cascade of oxidative stress and inflammatory signaling pathways, resulting in the increased production of inflammatory cytokines and chemokines. This, in turn, triggers an enhanced immune response characterized by the recruitment of immune cells, leading to a further increase in ROS (Reactive Oxygen Species) production catalyzed by NADPH oxidases (NOXs). Concurrently, there is prolonged activation of inflammatory, redox, and apoptosis signaling pathways. Ultimately, this cascade of events culminates in several adverse outcomes: a dysfunction in the immune response, increased cell apoptosis, disrupted mucosal homeostasis, and impaired intestinal barrier function, which can lead to mucosal damage”
- *Even in the introduction, exclude the description of specific objectives; Thank you for your comment. We have changed as follows: This comprehensive systematic review endeavors to scrutinize and synthesize the existing evidence regarding the impact and efficacy of various diets in the management and treatment of Ulcerative Colitis (UC) in both pediatric and adult populations.
- *Please use the "Flow diagram of study selection Prism" template; DONE
- *delete column "Study design" as being RCT is an inclusion criterion; DONE
- *As having UC is an inclusion criterion, there is no need to describe this fact in the results table, but it is essential to include the mean and SD of age; Thank you for your comment. Unfortunately the data requested by the reviewer is not available in all studies and in some of them it is also not expressed as mean and standard deviation. For this reason we have considered adding the type of study population (adults, children and/or adolescents).
- *What is "Elimination diet" by the author Candy (1995); DONE
- *In line 146, delete the term "Randomized Controlled Trials" and leave only the acronym RCT DONE
Reviewer 3 Report
The systematic review is well written, and the methodology appropriate. However, the main doubt of this reviewer is as follows: is it possible to extrapolate all these conclusions raised by authors from 14 clinical trials involving a total of 288 patients? Considering the complexity of a diet regimen and the pharmacological approach adopted for Ulcerative colitis patients, I believe that publications considered by authors are too few to allow clearcut conclusions. I am aware that the authors clearly state this point but, in my opinion, this is not enough and the paper does not add much to what is known about UC disease and dietary interventions.
Discussion and conclusions are verbose, with concepts that are often repeated along the two paragraphs.
Moreover, few points are not fully clear: it’s clear that systematic review considered all studies up to July 1st, 2023, but the starting date is not (all the clinical trials on UC published before July 1st 2023?). The terms “importance” and “certainty” reported in Table 2 and 3 should be clearly explained and reported in the Table’s legend. The authors did not explain the meaning of AID (Anti-inflammatory diet) and should report and comment in a clearer way the pharmacological treatments used in the clinical studies.
Author Response
AUTHORS: We would like to thank to Reviewer 3 its constructive comments that obviously have improved our manuscript.
Comment
The systematic review is well written, and the methodology appropriate. However, the main doubt of this reviewer is as follows: is it possible to extrapolate all these conclusions raised by authors from 14 clinical trials involving a total of 288 patients? Considering the complexity of a diet regimen and the pharmacological approach adopted for Ulcerative colitis patients, I believe that publications considered by authors are too few to allow clearcut conclusions. I am aware that the authors clearly state this point but, in my opinion, this is not enough and the paper does not add much to what is known about UC disease and dietary interventions. Thank you very much for your comments. We fully agree with your feedback. We have condensed and specified the discussion, and we have tempered the conclusions in line with the findings of this review. We respectfully disagree with the reviewer's comment suggesting that our article doesn't add much more than what has been published. In this regard, it's worth noting that we have updated and analyzed the available evidence, outlined strategic directions, and highlighted the weaknesses of the published articles. Without a doubt, while perhaps not for experts, for a less specialized audience, this article provides quality evidence beyond what has been previously published.
Comment
Discussion and conclusions are verbose, with concepts that are often repeated along the two paragraphs. We have shortened the discussion and conclusions, making them clearer and more concise. We hope they meet your approval.
Comment
- Moreover, few points are not fully clear: it’s clear that systematic review considered all studies up to July 1st, 2023, but the starting date is not (all the clinical trials on UC published before July 1st 2023?). Yes, in inclusion and exclusion criteria you can find this sentence: Primarily, articles published in the English and Spanish languages were considered, spanning from the inception of relevant databases to the date of July 1st, 2023.
- The terms “importance” and “certainty” reported in Table 2 and 3 should be clearly explained and reported in the Table’s legend. DONE
- The authors did not explain the meaning of AID (Anti-inflammatory diet) and should report and comment in a clearer way the pharmacological treatments used in the clinical studies. DONE
Round 2
Reviewer 3 Report
I appreciate very much the efforts done by authors to improve the manuscript according to reviewers' suggestion.
Manuscript deserves publication now.